# A Genome-Wide Association Study of Senegalese Sorghum Seedlings Responding to Pathotype 5 of *Sporisorium reilianum*

**DOI:** 10.3390/plants11212999

**Published:** 2022-11-07

**Authors:** Ezekiel Ahn, Coumba Fall, Louis K. Prom, Clint Magill

**Affiliations:** 1Department of Plant Pathology & Microbiology, Texas A&M University, College Station, TX 77843, USA; 2USDA-ARS Southern Plains Agricultural Research Center, College Station, TX 77845, USA

**Keywords:** sorghum, Senegalese accessions, seedling, headsmut, *Sporisorium reilianum*, pathotype 5, GWAS

## Abstract

*Sporisorium reilianum* is a fungal pathogen that causes head smut in sorghum. In addition to pathotypes (P) 1-4, P5 and P6 were identified recently. In this study, seedlings of Senegalese sorghum, comprising 163 accessions, were evaluated for response to *Sporisorium reilianum*. Teliospores of pathotype P5 of the pathogen in dilute agar were pipetted onto seedling shoots while still in soil, and inoculated seedlings were submerged under water at 4 days post-inoculation. Signs of infection (noticeable spots) on the first leaf were checked daily up to 6 days post submergence. A genome-wide association study (GWAS) was conducted using 193,727 single-nucleotide polymorphisms (SNPs) throughout the genome based on two types of phenotypic data: whether noticeable spots were shown or not and the average time for an observation of the spots across 163 accessions. When mapped back to the reference sorghum genome, most of the top candidate SNP loci were associated with plant defense or plant stress response-related genes. The identified SNP loci were associated with spot appearance in sorghum seedlings under flooding following inoculation with P5 of *Sporisorium reilianum*.

## 1. Introduction

Sorghum (*Sorghum bicolor* L. Moench) ranks fifth among the most grown cereals in the world [1] and is a valuable crop for addressing the global problems of climate change and growing population [2]. As a C4 grass, sorghum performs well under abiotic stresses like drought, seasonal flooding and low nutrient availability [3]. Sorghum serves as a staple food, feed and biofuel source throughout the world [2]. The United States is one of the leading countries in sorghum production worldwide [4], but the sorghum production industry is hampered by biotic stresses, especially plant diseases. The facultative biotrophic fungal pathogen *Sporisorium reilianum* (Kühn) Langdon & Fullerton causes head smut in sorghum. This basidiomycete pathogen has two formae speciales, ‘*reilianum*’ and ‘*zeae*’, that are, respectively, specialized on sorghum and maize. *S. reilianum f.* sp. *reilianum* infection on sorghum can lead to complete panicle loss under the successful penetration and colonization of the plant [5]. *S. reilianum* f. sp. *reilianum* is highly virulent on susceptible sorghum cultivars after the successful mating of two haploid yeast-like sporidia resulting from the germination of haploid soilborne teliospores. The sporidia carrying different mating types at both a and b loci recognize each other through a pheromone/pheromone receptor system and undergo anastomosis to form a dikaryotic filament that penetrates the plant surface; they then colonize the host. No noticeable symptoms of infection are detected until the invasion of the undifferentiated floral tissue, which will, at panicle emergence, result in the replacement of the inflorescence by sori containing black masses of teliospores [6]. Therefore, sorghum head smut might lead to a nearly complete yield loss. The use of resistant hybrids is the main cost-effective control measure deployed against *S. reilianum* f. sp. *reilianum* [7]. However, the occurrence of different races of the pathogen hampers the sustainable use of resistant cultivars. In 2011, Prom et al. [8] reported two new pathotypes (5 and 6) of *S. reilianum* f. sp. *reilianum* in South Texas that appeared on cultivars that were being grown to avoid pathotype 4, which is common in that area. In parallel, the knowledge of plant inhibitory compounds involved in defense mechanisms could allow a new approach for disease management that relies on the plant’s innate immune mechanisms [9].

In that sense, a previous genome-wide association study (GWAS) performed on the sorghum mini core lines for anthracnose, head smut and downy mildew revealed top candidate genes associated with single-nucleotide polymorphisms (SNPs), suggesting that leucine-rich repeat (LRR), tyrosine-kinases and zinc finger proteins are involved in sorghum defense against *S. reilianum*. The candidate genes themselves act in host defense aspects typical of quantitative trait loci (QTLs) and therefore are probably governed by minor genes [10].

In 1992, Craig and Frederiksen developed a seedling inoculation method for sorghum. In brief, sorghum seedlings were inoculated by infesting the vermiculite surrounding seedling epicotyls with teliospore cultures, and at 4 days post-inoculation, the inoculated seedlings were placed in test tubes filled with water to completely submerge the first leaf [11,12]. Based on symptoms on the first leaf blade, susceptible (showed brown or dark spots) and resistant (no spot) genotypes were differentiated [11,12]. A more recent work focused on the response of Senegalese sorghum seedlings to pathotype 5 of *S. reilianum*, showing that the defense-related genes, pathogen-induced chitinase and Pathogenesis-Related protein 10 (PR-10) were, to some degree, upregulated 24 h post-inoculation [11].

In the present study, seedlings of 163 lines were evaluated for response to the pathotype 5 (P5) of *S. reilianum* using the method described by Ahn et al. [11] in their previous study, which is a slightly modified version of the one from Craig and Frederiksen [12]. The seedlings were inoculated with a mixture of teliospores of *S. reilianum f.* sp. *reilianum* and sucrose–agar, then submerged in water four days post-inoculation before being evaluated for spot appearance up to six days after submergence. Based on the resulting phenotypes, GWAS was performed using the phenotypic data collected. In a previous study, the same accessions were analyzed for the population structure and linkage disequilibrium; germplasm diversity analysis showed low genetic diversity and slow linkage disequilibrium (LD) decay among the Senegalese accessions [13]. We hypothesized that candidate genes related to sorghum stress response could be identified. Here we report sorghum disease response and/or abiotic stress response-related genes associated with *S. reilianum* infection.

## 2. Results

### 2.1. Phenotypic Variation

ANOVA for the 163 accessions based on the two traits showed significant differences with *p* < 0.0001. Most of the Senegalese accessions showed spots (Figure 1), but 17 accessions were completely free of spots (Appendix A & Appendix A list the accessions ordered by average time of spot detection); spot appearance rates were between 0 to 100%. Average time of spot detection lies between 2- and 6-days post-submergence underwater.

No correlation was found between infection rate and time of appearance of symptoms (Pearson’s correlation = −0.12 with *p*-value = 0.14).

### 2.2. Genome-Wide Association Study

Based on susceptibility, 9 SNP loci passed the Bonferroni threshold when the GWAS was conducted, but 2 SNP loci did not pass the threshold of *p*-value < 0.05 with the *t*-test (Figure 2 and Table 1); hence, SNP loci S05_5262365 (tagged to Sobic.005G052300, which is associated with F-box domain) and S05_1285062 (near Sobic.005G014100, which is associated with MYB-like DNA-binding protein) were filtered out and are not listed in Table 1.

Based on average infection time, no SNP passed the Bonferroni threshold on GWAS.

A Manhattan plot was prepared showing the locations of SNP-detected QTLs associated with response to P5 of *S. reilianum* on the 10 chromosomes of *Sorghum bicolor* at 1-leaf stage. Bonferroni correction≈0.00000039 after filtering out SNPs with greater than 20% unknown alleles with minor allele frequency (MAF) below 5%.

When mapped back to the published reference sorghum genome, top candidate SNP loci were tagged or located nearest to genes that have previously reported roles in biotic or abiotic resistance/stress responses in most cases.

## 3. Discussion

As in a previous study, a possible correlation between infection rate and time of appearance of symptoms lacked statistical significance (Pearson’s correlation = −0.12 with *p*-value = 0.14) [11]. As in the previous study that used 36 accessions showing Pearson’s correlation = −0.08 with *p*-value = 0.66, the two traits might be weakly correlated.

As average time of spot appearance varied among the tested accessions, timing was expected to identify top candidate SNP loci associated with the trait through GWAS. However, the top candidate SNP loci did not pass additional t-tests to ensure that the SNP loci associated with timing of spot appearance were truly significant (data not shown, but raw data are available in Appendix A). Furthermore, GWAS based on spot appearance overall identified SNP loci with significant *p*-values. As head smut in sorghum is typically labeled as susceptible when even one sample shows symptoms, GWAS based on binary phenotypes (resistant or susceptible) was conducted, and the top candidate genes identified were associated with plant defense or plant stress response-related genes.

The SNP locus S04_55470704 tagged to F-box and leucine-rich repeat protein (Sobic.004G202700). F-box and leucine-rich repeat protein were listed as top candidate defense related genes in Senegalese accessions [13] and SAP accessions [14] to *Colletotrichum sublineola*. In sorghum mini core accessions, F-box and leucine-rich repeat proteins were top candidate genes to *C. sublineola*, *Peronosclerospora sorghi*, and *S. reilianum* [10]. The locus S04_6170007 matched (Sobic.004G273200) the Xyloglucan endotransglucosylase/Hydrolase protein 29-related. Xyloglucan endotransglucosylase/Hydrolases (XTHs) are a large family of enzymes, and the XTHs genes are known to be expressed in specific plant tissues [15]. XTHs have been reported for their important role in cell wall homeostasis for plant growth and development [16]. Indeed, these genes play an important role in cell wall reconstruction and stress resistance, particularly abiotic stresses. Thirty-five XTH genes have been reported in sorghum [17]. To the best of our knowledge, no biotic stress related function has been reported, suggesting that this SNP association could be due to switching to the flooded environment to which the seedlings were exposed during the experiment. Further investigation could help to elucidate this ambiguous finding. The SNP locus S04_65477983 (Sobic.004G319800) was located near the chloroplastic 10 kDa heat shock protein//20 kDa Chaperonin. Small heat shock proteins correspond to molecular chaperones that maintain the appropriate folding, trafficking and disaggregation of proteins under diverse abiotic stress conditions [18]. Nagaraju et al. [18] reported that sorghum small heat shock proteins were highly induced under abiotic stresses (heat, cold, drought and salt) and inferred their putative role in mediating environmental stress responses, in addition to plant development. These authors reported 17 genes corresponding to a large subfamily of subgroups and localized in the chloroplast. As suggested by our GWAS, these proteins might play a role under disease stress, though this would need stronger scientific evidence.

The SNP locus S03_73084958 tagged to a protein similar to the signal recognition particle 54 kDa protein 1 (Sobic.003G427400). The chloroplast signal recognition particle (cpSRP) 54 targets the light harvesting complex proteins to thylakoids and plays a critical role [19]. Moreover, two rice homologous impact chloroplast development and plant survival in rice [19]. These authors also suggested that these proteins might play distinct roles in transporting different chloroplast proteins into thylakoids through cpSRP-mediated pathway.

The SNP S01_65477983 was associated with a helix-loop-helix DNA-binding domain containing protein (Sobic.001G430000). The basic helix-loop-helix (bHLH) is a superfamily of transcription factors largely occurring in plants and animals, in addition to being the second largest transcription factor family in eukaryotes after MYB. Transcription factor members of the bHLH target genes are involved in physiological processes such as plant development and stress responses. In a previous study, Fan et al. [20] reported that the expression study of 12 sorghum bHLHs (SbbHLH) showed that they were affected under abiotic stress conditions (ultraviolet exposure, drought, cold, heat, salt and flooding). The inferred involvement of the bHLH in head smut seedling inoculation suggested by the present study suggests a potential role of these transcription factor genes in disease responses.

The SNP locus S02_55267519 seemed to be associated with ankyrin repeats (Sobic.002G174700), which are the largest family within the repeat proteins and are abundantly found in bacterial, archaeal, eukaryotic and viral genomes. These repeats are involved in protein-protein interactions to activate or suppress biological processes [21]. In their genome-wide association mapping of epi-cuticular wax (EW) genes in sorghum, Elango et al. [22] identified 37 putative genes associated with EW biosynthesis and transport, an ankyrin repeat was among the major EW biosynthetic genes. EW is known to reduce water loss and impart tolerance to several stresses, including pathogens. It could be inferred that the ankyrin repeats identified here might be associated with a response to the head smut pathogen inoculation.

Another SNP locus on the same chromosome, S02_55189900, was weakly similar to the transcription factor WRKY74 (Sobic.002G174300). WRKYs are among the largest transcription factor families in higher plants and regulate many biological processes, including abiotic and biotic stress responses. WRKY transcription factors control gene expression through a combination of positive or negative regulation [23]. Baillo et al. [23] reported that WRKY74 of sorghum (SbWRKY74) was upregulated under drought stress 12 h (peak) after seedling stress exposure. These authors, however, mapped the gene to chromosome 8. Nonetheless, as WRKY transcription factors are known to regulate biotic stress responses, the tag of a SNP provides a clue for their possible implication in pathogen response.

Apart from F-box and leucine-rich repeats, most of the annotated functions of the genes nearest to the SNP loci identified in the present study, have been reported for their role in abiotic stresses responses and/or plant development. This might indicate a possible involvement in pathogen infection, in which case they would be useful in marker-assisted selection against sorghum head smut at the seedling stage, but it seems more likely the genotypic data are related to the environmental switch to flooding under which the seedlings were exposed.

Accessions used as positive checks and their mock inoculation worked well (Appendix A). Additional sorghum accessions, BTx623, BTx635, and TAM428 did not show any spots, while BTx643, SC748-5, and PI609251 showed spots in one or more seedlings. Again, BTx635, PI609251, and TAM428 are often used as resistant checks and BTx623, BTx643, and SC748-5 are used as susceptible checks in syringe inoculation. As the results in this study did not match with typical syringe inoculation made in older plants, it is possible that cultivar responses differ by maturity, as has been demonstrated with anthracnose infection (22). Again, since the accessions used in this study were not screened using syringe inoculation, it might simply mean that the accessions showed different responses to P5.

## 4. Materials & Methods

### 4.1. Sorghum Seedlings

The 163 Senegalese sorghum accessions were provided by the USDA-ARS Plant Genetic Resources Conservation Unit. They were chosen since whole genome sequences were available and are quite diverse as shown in other analyses [24]. Along with the Senegalese sorghum accessions, BTx623, BTx635 (−), TAM428, BTx643 (+), SC748-5 and PI609251 were planted as well. BTx635 and BTx643 are widely used as negative and positive controls for syringe inoculation screening of head smut [8,12], but both accessions were resistant in the seedling inoculation method described by Craig and Frederiksen (spot appearance rate: BTx635 = 0% & BTx643 = 10%) [12].

Seedlings were grown as described by Ahn et al. [11]. In brief, seeds were sown in cells of 40 square plugged flats (V ≈ 175 cm^3^) (The HC companies, Twinsburg, OH, USA) filled with Metro Mix 200 (Sun Gro Horticulture, Agawam, MA, USA). The seedlings were grown at 23℃ and 65% relative humidity under LED light for around 12 hours a day in the insectary room within the Plant Pathology and Microbiology (PLPM) building at Texas A&M University (College Station, TX, USA).

### 4.2. Seedling Inoculation

Harvested teliospores collected directly from the symptomatic plants inoculated with isolates of P5 of *S. reilianum* were provided by Louis K. Prom (USDA-ARS Southern Plains Agricultural Research Center). The inoculum was prepared following a modified version of that described by Craig and Frederiksen [12]. 0.2g of teliospores were suspended in 15 mL of sterile distilled water within a microcentrifuge tube before being precipitated by centrifugation at 500× *g* for 10 s. The supernatant was then poured off before the teliospores were resuspended in 15 mL of sterile distilled water. The process was repeated three times. After the third wash–resuspension step, 0.5 mL of the suspension (approximately 16 × 10^6^ teliospores) were used to inoculate an Erlenmeyer flask containing 50 mL of sucrose–agar medium (3% sucrose, 0.25% agar, *w/v*, adjusted to pH = 3.8 using lactic acid). The flask was then incubated on a rotary shaker at 100 rpm at room temperature for 7 to 10 days. Following the incubation period, at least 10 seedlings of each accession were individually inoculated with 1 mL of the resulting culture. BTx635, PI609251 and TAM428 are often used as resistant checks and BTx623, BTx643, and SC748-5 are used as susceptible checks. The inoculated seedlings were surrounded with vermiculite, then the plants were grown in the same conditions as previously described. Four days post-inoculation, the below ground part of each seedling was separated from its upper part by cutting at approximately 0.5 cm below the base of the stem. The seedling was then surface washed with tap water before being submerged in distilled water contained in a 45 mL plastic test tube. Each seedling was checked daily for spot appearance and percentage area coverage, up to six days, and results were recorded. The inoculations for each cultivar were repeated at least 3 times (total number of seedlings tested per accession = 10–32 as shown in Appendix A).

### 4.3. Statistical Analysis and GWAS

A Pearson correlation coefficient was calculated to determine if there was a correlation (Type 1 error rate α = 0.05) between the time of spot appearance and the percentage of leaf covered, using JMP Pro 15 (SAS Institute, Cary, NC, USA). Analysis of variance (ANOVA) for the 163 accessions based on the two parameters were conducted.

The SNP data used for the GWAS were from an integrated sorghum SNPs dataset obtained using Genotyping by Sequencing (GBS) based on the sorghum reference genome version 3.1.1 [25,26,27,28]. Missing data were imputed with Beagle 4.1 [29]. A mixed linear model (MLM) association analysis was conducted using TASSEL version 5.2.80 [30] based on whether spots appeared (binary: present or absent), regardless of the leaf area covered, i.e., plants showing spots considered susceptible and the ones without spots resistant. SNPs with more than 20% unknown alleles were removed to decrease false associations, followed by the removal of those with minor allele frequencies (MAF). SNPs associated with responses to *S. reilianum* inoculation were tracked to identify their specific location using the partially annotated sorghum genome sequence version 3.1.1, accessed through the JGI Phytozome 13 website [31]. The spot appearance rating for all accessions with either of the two prevalent SNP bases was determined and verified to differ significantly (*p* < 0.05) based on Student’s T-test using JMP Pro 15 (SAS Institute, Cary, NC, USA) [32]. The top candidate genes surpassed the Bonferroni test as well as those with a *p*-value lower than the threshold (95%) are reported here.

## Figures and Tables

**Figure 1 plants-11-02999-f001:**
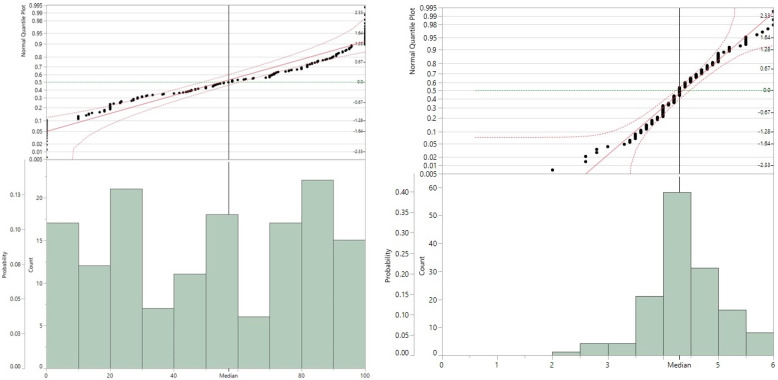
Phenotype distribution of infection rate (**left**) and average time for spot detection (**right**) among 163 sorghum accessions.

**Figure 2 plants-11-02999-f002:**
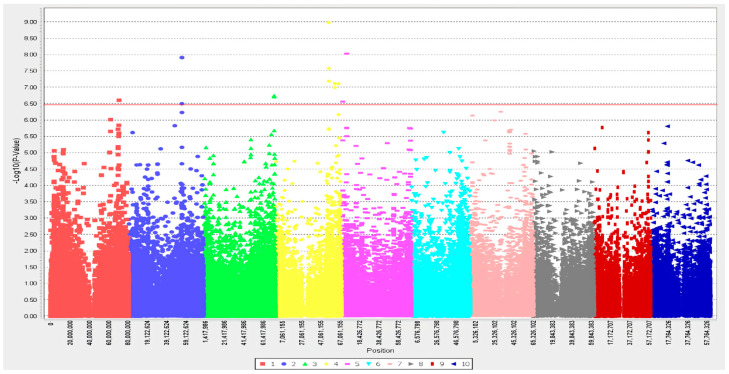
The genome-wide association for spot appearance to *S. reilianum* in Senegalese sorghum seedlings. Different color codes represent chromosome 1 through 10. Red line indicates Bonferroni correction≈0.00000039. SNPs above Bonferroni correction were considered as top candidate SNPs.

**Table 1 plants-11-02999-t001:** Annotated genes nearest to the most significant SNPs associated with spot appearance based on resistance/susceptibility to 1-leaf stage seedlings.

Chr.	Location	Candidate Gene and Function	Base Pairs	SNP Base %	TASSEL *p*-Value	% of Susceptible (Spot Present) Accession/SNP	Marker R^2^
4	55470704 and 4 more within 582 bp	Sobic.004G202700 F-box and leucine-rich repeat protein	0	G:12% T:88%	0.00000000105	G:100% T:91%	0.3
2	55267519 and 1 more within 7 bp	Sobic.002G174700 Ankyrin repeats	0	C:74% G:26%	0.0000000124	C:97% G:83%	0.27
4	61700071 and 2 more within 847 bp	Sobic.004G273200 Xyloglucan endotransglucosylase/Hydrolase protein 29-related	0	A:26% T:74%	0.000000077	A:80% T:96%	0.23
4	65477983	Sobic.004G319800 10 kDA heat shock protein//20 kDA Chaperonin, Chloroplastic	719	A:92% T:8%	0.0000000785	A:95% T:46%	0.23
3	73084958 and 1 more within 13 bp	Sobic.003G427400 Similar to signal recognition particle 54 kDa protein 1	0	C:6% G:94%	0.000000182	C:100% G:91%	0.22
1	70919811	Sobic.001G430000 Similar to Helix-loop-helix DNA-binding domain containing protein, expressed	1768	C:45% T:55%	0.000000249	C:85% T:98%	0.21
2	55189900	Sobic.002G174300 Weakly similar to Transcription factor WRKY74	0	C:83% T:17%	0.000000319	C:96% T:77%	0.21

The distance in base pairs to the nearest genes and *p*-value are listed. Two major alleles were calculated to verify significant differences for scores based on Student’s t-test, and SNPs failed to show differences were not considered as top candidates. All SNPs in this table passed Bonferroni correction.

## Data Availability

All the data generated in this study can be found in Appendix A.

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
