# Peer review of "A Genome-Wide Association Study of Senegalese Sorghum Seedlings Responding to Pathotype 5 of *Sporisorium reilianum"

_plants, 2022, doi:10.3390/plants11212999_

Round 1
Author Response
Please see uploaded word document

Reviewer 2 Report
This manuscript is well written and easy to follow and understand. However, due to the authors’ previous publication, I found a few paragraphs are the same as those in their publications. To avoid repetition, I recommend rewriting those sentences in the manuscript.
In the introduction, Page 2, lone 46: before introducing pathotypes 5 and 6, please give a little bit of the introduction of pathotype 1 to 4 and its distribution and virulence so that readers can have a brief overview of different pathotypes of S. reilianum.
Lines 57-63: completely repeat to another publication "Ahn, E., Prom, L. K., Fall, C., & Magill, C. (2022). Response of Senegalese Sorghum Seedlings to Pathotype 5 of Sporisorium reilianum. Crops, 2(2), 142-153." Please rephrase this paragraph.
Materials and Methods:
Line 79: Are the 163 accessions diversity panel?
Lines 81-83: This statement conflicts with that in the publication "Ahn, E., Prom, L. K., Fall, C., & Magill, C. (2022). Response of Senegalese Sorghum Seedlings to Pathotype 5 of Sporisorium reilianum. Crops, 2(2), 142-153.". Please double check whether BT635 and BT643 were resistant or not tested by the seedling inoculation method.
Line 92: Are those teliospores collected directly from the symptomatic plants?
Line 94: Do you know the concentration of the inoculum?
In the results, the disease screening among 163 accessions was only conducted one time without any replications, I am not sure how reliable the phenotypic data is. In addition, the data analysis is insufficient, superficial without depth. I think the results need to be rewritten.
Line 132: Can you show some pictures of the highly susceptible plants and the immune plants?
Line 138: Table 1 is too big, nobody really cares about the disease response on each line, therefore it's better to move Table 1 to the supplementary file. Instead of the table, you could use a graph to show how many lines were highly susceptible (HS), S, MR, and R.
Line 139: Have you run ANOVA for the 163 accessions based on the two parameters? I am wondering if any significant genotypic variation was detected among the 163 accessions?
Line 154: Please add a line for the threshold in the Manhattan plot.
Discussions:
Lines 171-172: Please explain why “it is speculated that sorghum responses differed by maturity”? Do you have any evidence to support your statement?
Author Response
See uploaded word document

Round 2
Reviewer 2 Report
I accept the MS in its current version
